# Improving Breast Cancer Detection and Diagnosis through Semantic Segmentation Using the Unet3+ Deep Learning Framework

**DOI:** 10.3390/biomedicines11061536

**Published:** 2023-05-25

**Authors:** Taukir Alam, Wei-Chung Shia, Fang-Rong Hsu, Taimoor Hassan

**Affiliations:** 1Department of Information Engineering and Computer Science, Feng Chia University, Taichung 407, Taiwan; taukir.alam007@gmail.com; 2Molecular Medicine Laboratory, Department of Research, Changhua Christian Hospital, Changhua 500, Taiwan; 3Institute of Translational Medicine and New Drug Development, China Medical University, Taichung 404333, Taiwan; taimoorhassan408.th@gmail.com

**Keywords:** deep learning, convolution network, semantic segmentation, ultrasound

## Abstract

We present an analysis and evaluation of breast cancer detection and diagnosis using segmentation models. We used an advanced semantic segmentation method and a deep convolutional neural network to identify the Breast Imaging Reporting and Data System (BI-RADS) lexicon for breast ultrasound images. To improve the segmentation results, we used six models to analyse 309 patients, including 151 benign and 158 malignant tumour images. We compared the Unet3+ architecture with several other models, such as FCN, Unet, SegNet, DeeplabV3+ and pspNet. The Unet3+ model is a state-of-the-art, semantic segmentation architecture that showed optimal performance with an average accuracy of 82.53% and an average intersection over union (IU) of 52.57%. The weighted IU was found to be 89.14% with a global accuracy of 90.99%. The application of these types of segmentation models to the detection and diagnosis of breast cancer provides remarkable results. Our proposed method has the potential to provide a more accurate and objective diagnosis of breast cancer, leading to improved patient outcomes.

## 1. Introduction

Breast cancer is a major threat to women worldwide, and early detection and diagnosis are critical to improving cure rates and reducing mortality [1]. Breast cancer has complex and diverse causes, including environmental factors, hormone secretion, and life stress [2]. At the molecular level, breast cancer is a highly heterogeneous disease that involves DNA damage, genetic mutations (such as *BRCA* gene mutations), activation of hormone receptors (progesterone and estrogen receptors), and expression of human epidermal growth factor receptor 2 (HER2). Early detection plays a crucial role in reducing breast cancer mortality rates in the long term. Identifying early stage breast cancer is essential for achieving optimal prognosis [3], with imaging examinations being an important means for diagnosis in the early stages.

Breast ultrasound is a non-radiation clinical modality that is well tolerated by patients and is widely used in the diagnosis of breast cancer [4,5]. However, its accuracy depends on the expertise of the clinician, which can lead to misdiagnosis [6]. Other diagnostic modalities, such as magnetic resonance imaging (MRI), mammography, computed tomography, digital breast tomosynthesis and positron emission topography, are also used, but have technical limitations [7]. To address these challenges, computer-aided diagnosis (CAD) algorithms, based on advanced artificial intelligence (AI) technology, can assist medical staff in accurately interpreting images. This tool has the potential to improve patient outcomes by guiding clinicians to correct diagnoses [8,9]. Breast ultrasound classification is crucial for distinguishing between benign and malignant tumours. Segmentation, which divides an image into regions to identify regions of interest (ROI), is a critical notion in image processing. However, segmentation can be problematic and lead to misdiagnosis, for example, when pectoral muscles are present but not part of the breast tissue [10,11]. Therefore, it’s important to compare identified cancer ultrasound images with the standard Breast Imaging Reporting and Data System (BI-RADS) from a clinical perspective. BI-RADS provides standardised terms, known as a lexicon, that describe the characteristics of breast masses and effectively differentiate between benign and malignant tumours [12].

The ultrasound lexicon has several advanced features for inspecting and correlating breast cancer images based on: (1) breast composition (homogeneous fat, homogeneous fibro-glandular or heterogeneous); (2) mass shape (irregular, round, oval), margins (not-circumscribed, circumscribed, indistinct and speculated), and orientation (non-parallel or parallel); (3) calcifications location (outside mass within mass intraductal); and (4) associated features (architectural distortion, edema, skin retraction, etc.). It serves as a guide for radiologists/physicians to compare tumour images with high accuracy [13,14]. Recently developed deep learning systems have excellent accuracy in mass segmentation in breast ultrasound images using the Artificial Neural Network (ANN) [15], making it an advanced tool for the classification and analysis of breast cancer [16]. 

Our study focuses on improving breast cancer detection in ultrasound images using the Unet3+ architecture for semantic segmentation modelling. The Unet3+ architecture is a robust semantic network that shows promising results in medical imaging applications by addressing skip connections to improve the accuracy of Unet segmentation [17]. Skip connections allow information transfer between encoder and decoder layers of a UNET architecture to improve segmentation performance by more effectively capturing multi-scale contextual information [18]. We aim to improve the accuracy and efficiency of the Unet3+ model in detecting multiple malignancy-related image features simultaneously. Table 1 shows the different approaches used for pre-processing, segmentation and classification.

## 2. Materials and Methods

The paper presents an algorithmic framework, as illustrated in Figure 1, for analysing medical images of 309 patients. The dataset comprises 151 benign and 158 malignant images that are pre-processed to remove irrelevant information and resized for normalization. Semantic segmentation is then applied to classify each pixel based on its target feature, and precisely outline each object. The study compares the performance of Unet3+ with other models, such as FCN, Unet, Segnet, DeeplabV3 and pspNet, using metrics such as overlap, with ground truth and global accuracy. Finally, the study reports mean accuracy, mean IoU (Intersection over Union), weighted IoU and weighted F1 score for all models including Unet3+.

### 2.1. Image Data Collection Procedures

This is a retrospective, cross-sectional study approved by the institutional review board (IRB) of Changhua Christian Hospital, Taiwan (No. 181235). Informed consent was waived, and the ethics committee reviewed all experimental methods to ensure compliance with appropriate standards and the Helsinki Declaration. Participants were individuals between 35 and 75 years of age whose tumours had been diagnosed as benign or malignant by fine-needle cytology, core-needle biopsy or open biopsy. Detailed data on patient treatment, such as therapy, histology and radiography, were also collected. Breast ultrasound images were obtained using a GE Voluson 700 (GE Healthcare, Zipf, Austria). For each participant, at least two, different, scan-plane angles were acquired with each image providing the full screen of the scan plane in RGB mode at a resolution of 960 × 720 pixels.

### 2.2. Image Data Preprocessing 

To pre-process the breast ultrasound images, we first removed irrelevant information, such as manufacturer’s labels, directional markings and text fields, which could interfere with image interpretation and lead to incorrect results during noise reduction. This step ensured a clear and focused image [28]. Next, the images were normalised and transformed to a consistent size and format for efficient processing by machine learning algorithms. As different resolutions and aspect ratios could affect the accuracy of the analysis, it was crucial to resize the images to a standard size and aspect ratio. Finally, we validated the data before feeding it into the algorithm. 

### 2.3. Semantic Segmentation of Images

Image segmentation is the process of dividing an image into memory blocks that are equally related, with no deeper meaning within the divided area. This operation typically relies on low-level representations, such as colour, texture, and boundary smoothness, to cluster and cut output regions or segments. However, this method does not identify regions that belong to the same or related categories. Semantic segmentation, on the other hand, classifies each pixel in an image based on its target feature. The result is a precise outline of each object or feature to understand the overall meaning of the image. Semantic segmentation is crucial to understanding images because it makes them more meaningful and easier to analyse. In essence, semantic segmentation can be viewed as a pixel-by-pixel classification problem, where pixels or super-pixels are classified into different categories for breast cancer detection and segmentation [29].

### 2.4. Training Protocol and Infrastructure

The learning rate was set to 0.01 with a maximum epoch of 100 and a batch size of two. Gaussian blur was applied to the images to reduce noise and smooth edges, while random rotation up to 90° and elastic transformation were used for distortion correction. The image size was fixed at 360 × 360 pixels with pixel normalisation using mean and standard deviation. 

The computational platform used in this study consisted of an Intel Core i5-11400F (2.6 GHz hexa-core with up to 4.4 GHz turbo boost and 12 MB cache) (Intel Corp., Santa Clara, CA, USA), NVIDIA RTX3060 graphics card with 12 GB video RAM (Nvidia Corp., San Jose, CA, USA) and Compute Unified Device Architecture (CUDA) version 11.2 (Nvidia Corp., San Jose, CA, USA). This setup provided an accelerated computational environment through the use of the NVIDIA processing unit. All programmes related to this study were implemented using the Pytorch framework version 3.7 (PyTorch, San Francisco, CA, USA).

### 2.5. Semantic Segmentation Network Model

The Unet3+ [17] architecture is an advanced version of the original Unet architecture originally designed for biomedical image segmentation. It incorporates several features that improve the accuracy of the semantic segmentation results. The architecture consists of two networks: an encoder network and a decoder network. The encoder network reduces the spatial dimensions of an input image using convolutional and pooling layers, while the decoder network upsamples the reduced feature map to its original size using convolutional and upsampling layers.

Figure 2 is the structural diagram of the Unet model and its related variants (such as Unet++, Unet3+). When compared to Unet, a notable aspect of Unet3+ is the dense connections between the encoder and decoder layers within a block, which connect all subsequent blocks in the network. This facilitates the flow of information between layers, thereby improving the accuracy of semantic segmentation. In addition, residual links are used to connect distant layers in the network to allow information propagation across multiple layers, further improving performance [30,31].

Both Unet and Unet++ do not explore enough information from full scale, resulting in the inability to explicitly determine the position and boundary of an organ. However, each decoder layer in Unet3+ contains smaller and equal scale feature maps from the encoder, as well as larger scale feature maps from the decoder. This allows fine-grained detail and coarse-grained semantics to be captured at full scale. Unet 3+ produces a side output from each decoder stage that is supervised by the ground truth. To achieve deep supervision, the last layer of each decoder stage undergoes a simple 3 × 3 convolution followed by bilinear upsampling and a sigmoid function.

### 2.6. Compared to Related Semantic Segmentation Network Model

In our study, we investigated six related segmentation models. We compared the performance of breast cancer segmentation on medical images, including SegNet [11], DeepLabv3+ [32], Fully Convolutional Neural Networks (FCNs) [33], pspNet (Pyramid Scene Parsing Network) [34] and EfficientNet. SegNet is a deep learning model that excels at semantic segmentation tasks due to its encoder–decoder architecture that extracts high-level features while preserving spatial information. DeepLabv3+ extends DeepLabv3 by adding an encoder–decoder structure. The encoder module processes multiscale contextual information through extended convolution at multiple scales, while the decoder module refines segmentation results along object boundaries. The FCN model is a well-known classification network model based on the VGG16 architecture and has shown potential in identifying and localising breast tumours [35]. The Pyramid Scene Parsing Network (PspNet) [34] and EfficientNet are utilised for precise breast cancer detection in medical images. PspNet incorporates a pyramid pooling module to gather contextual information at various scales, while EfficientNet employs scaling and compound scaling techniques to attain superior accuracy with fewer parameters than other models.

## 3. An Analysis and Evaluation

The performance of the semantic segmentation was evaluated by measuring the overlap with the ground-truth image dataset. This evaluation included global accuracy, mean accuracy, mean/frequency weighted intersection over union (IU) and mean F1 score. Indicators, such as accuracy, precision, recall and comprehensive evaluation index (F-measure), were used to evaluate the proposed method. The number of predicted true positives (TP), false positives (FP), true negatives (TN) and false negatives (FN) were also considered.

Accuracy is a measure of the probability of correctly classifying samples. The formula for accuracy is (TP + TN)/(TP + FP + TN + FN). Precision measures how well a sample predicts a particular category. The precision formula is TP/(TP + FP). Recall measures the proportion of true positives that were correctly identified in the sample. The recall formula is TP/(TP + FN). F-measure, also known as F-score, considers both precision and recall to evaluate the accuracy of a model in binary classification problems. The formula for the F-measure is 2 × ((Precision × Recall)/(Precision + Recall)). Global Accuracy refers to the percentage of pixels that are correctly classified regardless of category, and can be calculated quickly and easily using scoring criteria. Mean Accuracy or Average Class Accuracy calculates the ratio of correctly classified pixels to the total number of pixels in each class. This metric can be misleading if one category has significantly more labelled pixels than others. Intersection over union (IoU), also known as the Jaccard similarity coefficient, evaluates semantic segmentation by calculating the correlation between correctly classified pixels and the ratio of ground truth to predicted pixels. IoU values range from 0 to 1 with higher coefficients indicating greater sample similarity.

## 4. Results

Figure 3 shows the semantic segmentation performance (Metrics) of the six models used in this study, including Unet3+, Unet, FCN-32s, SegNet, pspNet and DeeplabV3+. From Figure 3, we can see that Unet3+ had the highest global accuracy with better mean accuracy, mean IoU and weighted F1 score. Specifically, Unet3+ achieved 0.90 for mean accuracy, 0.82 for mean IoU and 0.52 for mean weighted IoU, while its weighted F1 score was found to be 0.83 with superior Parmas (KB) compared to other models. Although DeepLabV3+ had a higher global accuracy of 0.91 than other models, our analysis showed that Unet3+ outperformed it in terms of overall performance, based on metrics such as mean accuracy, mean IoU, weighted F1-score and Parmas (KB). Figure 4 shows the results presented separately for each model and indicator item, to facilitate comparison between models. Table 2 lists the original data.

## 5. Discussion

In this study, we employed the Unet3+ deep learning architecture to accurately segment the malignant BI-RADS lexicon feature of breast cancer in medical images. Our approach successfully addressed the challenge of identifying the small and irregularly shaped malignant features on a tumour, which is a difficult task in medical image analysis. The segmentation map clearly demonstrates that our use of learning networks and fully connected networks resulted in an improved BI-RADS lexical recognition compared to other models such as Unet, FCN 32s, SegNet, DeepLabV3+, and pspNet. Notably, the Unet3+ model outperformed all other models tested when applied to the test set.

Figure 5 shows the results of semantic segmentation of breast ultrasound images using the model used in this study. In Figure 5, we simultaneously allowed six models to perform semantic segmentation on ten randomly selected images from the image dataset and provided their ground truth as a comparison for the segmentation results. FCN-32 and Unet produced good segmentation results, while SegNet performed poorly in comparison. These findings suggest that the classification or recognition performance of a well-trained deep learning model, may decline to some extent when applied to datasets with varied sizes or characteristics. In comparing FCN-32s and deepLab3+ with ground truth, DeepLabV3+ showed improved recognition of shadowing features. Similarly, in comparing SegNet and pspNet with ground truth, pspNet demonstrated an enhanced recognition of shadowing features due to its emphasis on obtaining multi-scale contextual information through the pyramid pooling module. This allowed for effective integration of information at multiple scales and established a thorough comprehension of complex scenes and objects, resulting in more accurate segmentation findings aligned with the ground truth.

DeepLabV3+ also demonstrated the competitive performance when it came to matching the ground truth. DeepLabV3+ was able to capture multi-scale information without downsampling, maintaining spatial features. The decoder module, in conjunction with skip connections, aided in the recovery of lost spatial data, which enhanched segmentation accuracy even further. 

The Unet architecture adequately reflected the ground reality and demonstrated ease of use and good performance in semantic segmentation tasks. The incorporation of skip links between the encoder and decoder facilitated the merging of low-level and high-level characteristics, resulting in a more precise object localization. While Unet’s performance was not as strong as that of Unet3+, pspNet, or DeepLabV3+, it still produced segmentations that closely approximated the ground truth.

The pspNet framework, despite incorporating the improved Unet3+ module, did not perform satisfactorily. It encountered issues with pixel category classification, leading to significant deviations from the ground truth in twenty-two segments. In contrast, the Unet3+ model utilized the redesigned skip connections and combined multiple scale features through multi-scale deep supervision. This theoretically enhances its ability to acquire and fuse image features while computing pixels for better segmentation results than the Unet model. However, experimental outcomes failed to demonstrate any improvement over the Unet model, as there was a noticeable loss of detail observed in the test set images [36,37,38].

Figure 6 illustrates the loss function curve in training process of the Unet3+ model. The following metrics are shown: (a) Loss_Train, which represents the loss function during training; (b) Loss_eval, which is the value of the loss function computed on the validation set during model evaluation; (c) MIOU_Train, which is Mean Intersection over Union (MIOU) computed on the training set during model training; and (d) MIOU_eval, which refers to Mean Intersection over Union (MI-OU) computed on the validation set during model evaluation. The loss curve for the training set gradually decreases with an increase in the number of epochs. It is important to note that Unet3+ belongs to the deep supervision model and has five outputs calculating the loss, which requires dividing by five to obtain a loss of 0.269 from 1.347. However, there is significant fluctuation in the loss function of the validation set, reaching a value of 0.252 at epoch = 100 from 1.281/5 = 0.252. There exists a considerable difference between MIOU_train (0.894) and MIOU_eval (0.453), representing training and validation sets, respectively, in terms of numerical values. During the training phase, the Unet3+ model’s loss and MIOU curves exhibit volatility with significant numerical disparities between the training and validation sets.

## 6. Conclusions

This study demonstrates the effectiveness of the Unet3+ model, a modified version of the Unet architecture, in accurately segmenting breast tumours from mammogram images. By incorporating dense skip connections and attention gates, this model can capture multi-scale features and focus on important regions of the image. These improvements lead to more accurate tumour segmentation and highlight the potential for deep learning models to improve breast cancer diagnosis and treatment. The Unet3+ model provides a promising platform for further research in this area and, after refinement and optimisation, has the potential to be a reliable algorithm for radiologists and clinicians in the detection and diagnosis of breast cancer.

To fully utilize the potential of the model in clinical practice, future work should focus on enhancing and optimizing it. This can be achieved by investigating the application of transfer learning to adapt the model to various medical imaging modalities, including clinical data that can improve segmentation accuracy. Additionally, verifying the performance of the model on larger datasets with a wider range of images is necessary. Integrating the Unet3+ model with decision support systems would provide doctors access to more thorough analyses of medical imaging. This integration will enhance breast cancer detection and management capabilities.

## Figures and Tables

**Figure 1 biomedicines-11-01536-f001:**
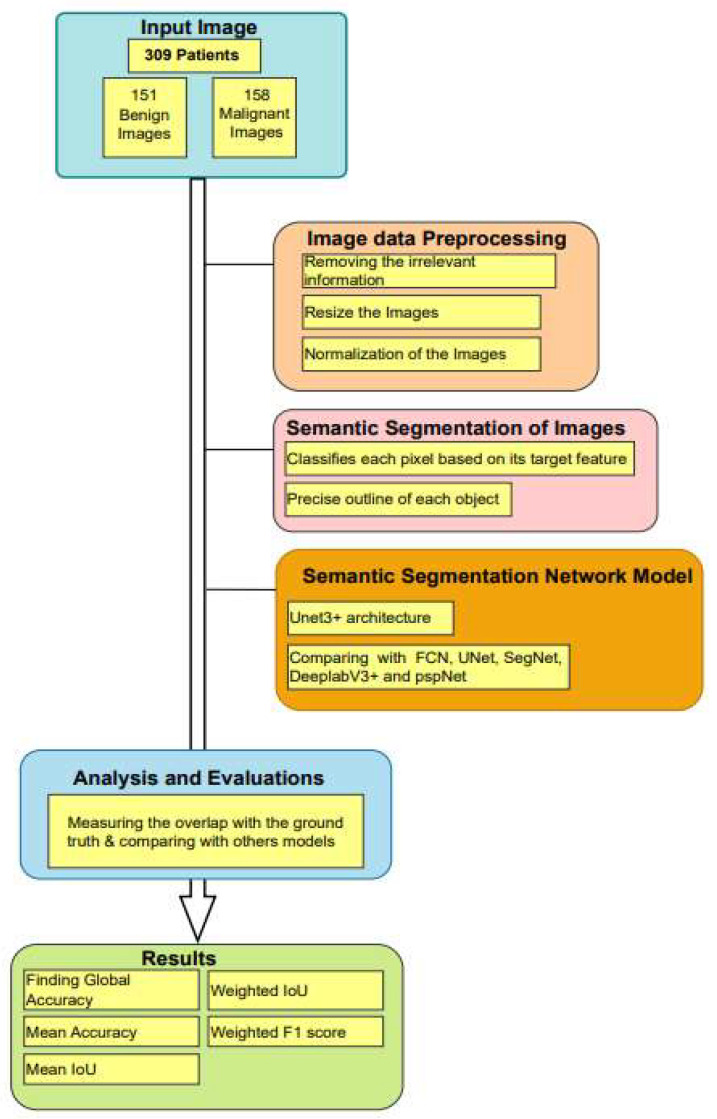
The overall algorithm and framework of this paper.

**Figure 2 biomedicines-11-01536-f002:**
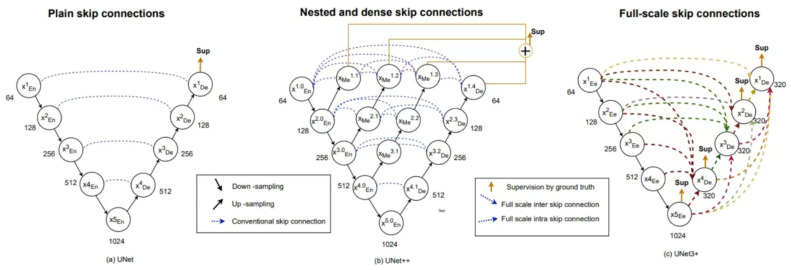
Structural diagram of Unet and its related models. (**a**) Unet, (**b**)Middle Unet++, (**c**) Unet3+.

**Figure 3 biomedicines-11-01536-f003:**
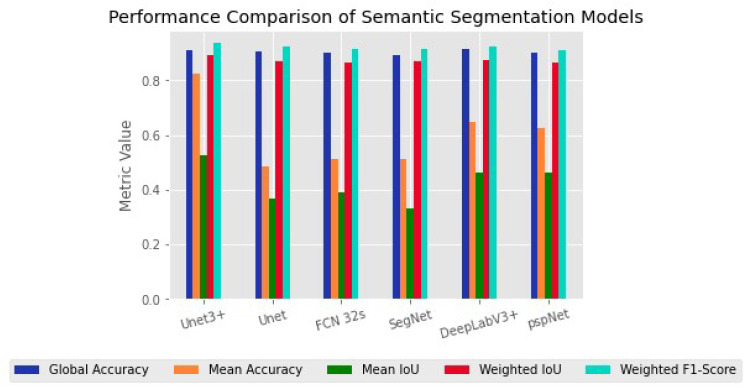
Performance Comparison of Semantic Segmentation Model.

**Figure 4 biomedicines-11-01536-f004:**
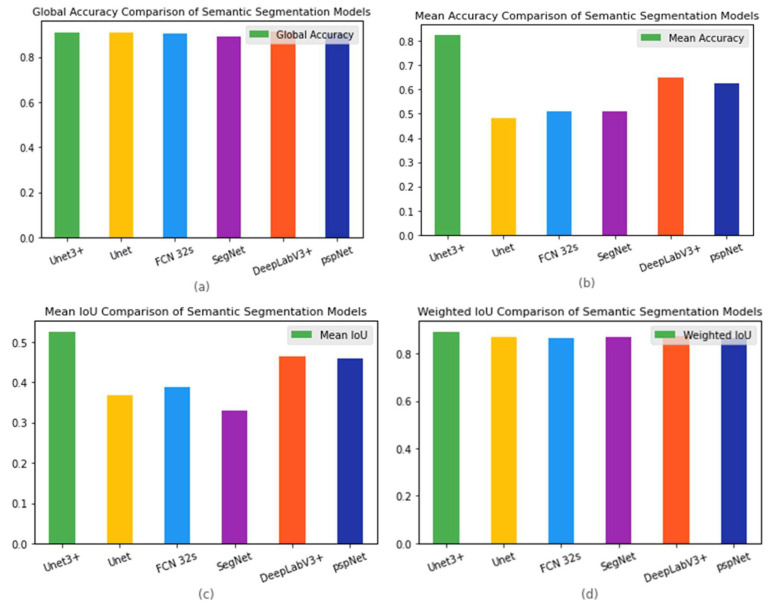
The performance comparisons of all semantic segmentation models using in this study are displayed graphically. Green: Unet3+, yellow: Unet, blue: FCN-32s, purple: SegNet, orange: Deeplab V3+, deep blue: pspNet. (**a**) Global accuracy, (**b**) Mean Accuracy, (**c**) Mean IoU, (**d**) Weighted IoU (Intersection of Union), (**e**) Weighted F1-score and (**f**) Parmas (in KB).

**Figure 5 biomedicines-11-01536-f005:**
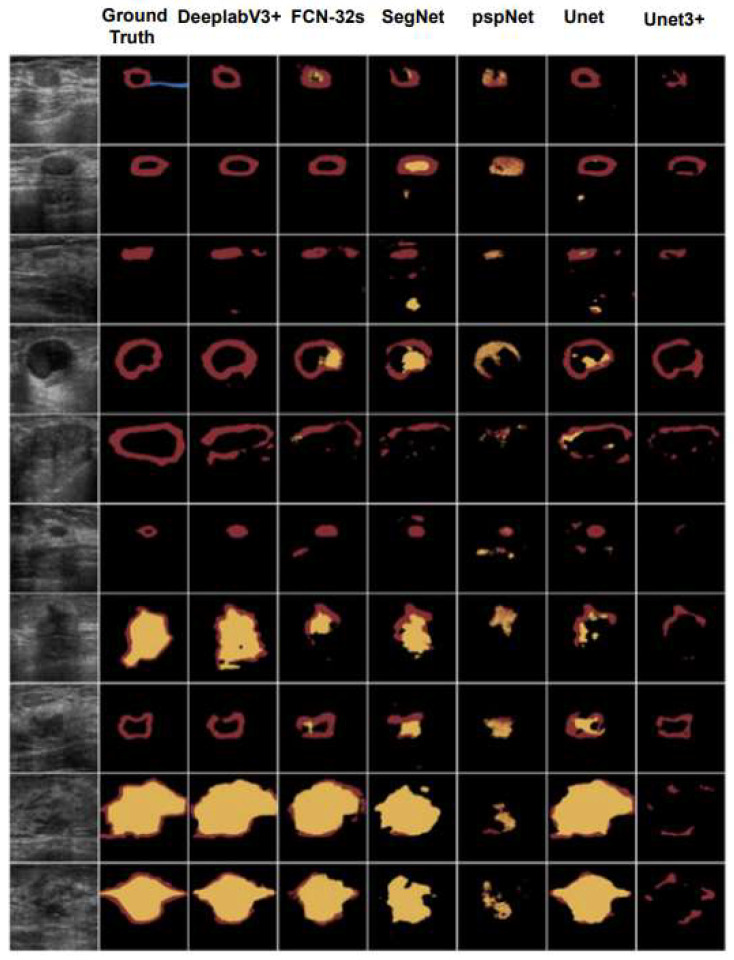
The sample of semantic segmentation visualisation for the malignant tumor ultrasound, and comparing Unet3+ with DeeplabV3+, FCN-32s, SegNet, pspNet and Unet.

**Figure 6 biomedicines-11-01536-f006:**
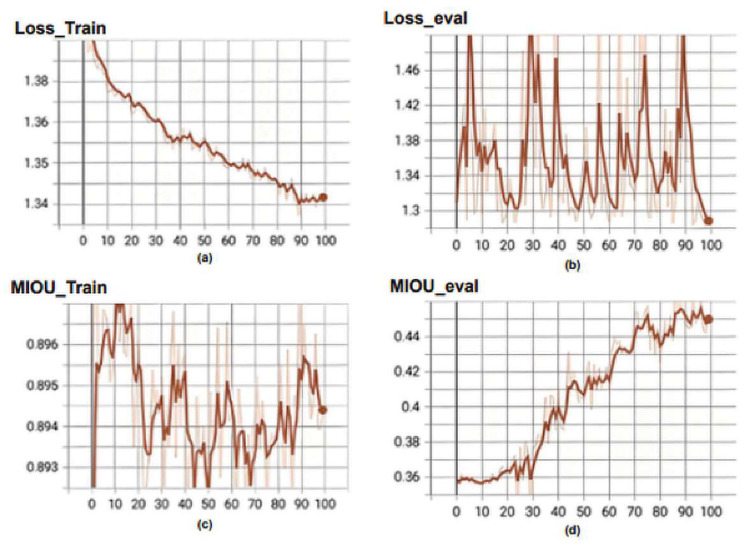
Training curve and training process of the Unet3+. (**a**) Loss_Train, (**b**) Loss_eval, (**c**) MIOU_Train and (**d**) MIOU_eval.

**Table 1 biomedicines-11-01536-t001:** Summary related semantic segmentation studies with their different modules, data used and results.

Authors	Topic Area	Preprocessing Technique	Segmentation Technique	Dataset	Final Results
S.Punitha et al. [19]	Segmentation of benign and malignant breast cancers utilizing an optimal region growth technique	Manual Cropping and Gaussian Filtering	Dragonfly Optimized Region Growing	DDSM Images in CC and MLO views	Sensitivity: 98.1% Specificity: 97.8%
Rahimeh et al. [20]	The classification of benign and malignant breast cancers is based on area growth and CNN segmentation.	Median Filtering	Region Growing and CNN optimized using GA	MIAS and DDSM	Accuracy: 96.47%
Arden et al. [21]	Law’s textural energy measure and neural networks are used to classify mammograms.	N/A	Cropping	MIAS	Accuracy: 93.90%
Celia et al. [22]	Breast mass identification in digitized mammograms by computer.	Iris filter	Adaptive Threshold method	Images from hospitals of the health district of Santiago deCompostela, Spain	Sensitivity of 88% and 94% at 1.02 false positives per image
Bhagwati et al. [23]	Mammography feature analysis and mass detection in breast cancer images.	Homomorphic filter	Region Growing	DDSM	Accuracy: 95.6%
Danilo et al. [24]	Breast cancer identification and segmentation in mammograms using wavelet analysis and genetic algorithms. Compute Methods.	Wiener Filter	Hammouche’s Computational algorithm using wavelet and Genetic Algorithm	DDSM-Images in CC and MLO Views	AOM: 79.2 ± 8%
Zhili C et al. [25]	Mammographic microcalcification cluster topological modelling and classification.	Manual segmentation	Topological features	MIAS and DDSM	Accuracy: 96%
Qaisar et al. [26]	Breast mass segmentation in a 4-stage multiscale system employing region-based and edge-based approaches.	CLAHE, Gaussian Filtering	Multiscale feature Fusion and Maximum a posteriori method	Mini MIAS and DDSM	AOM: 90%
Kanchan et al. [27]	Digital mammography can identify breast cancer.	Median Filtering	Fuzzy-C-means and thresholding technique	Mini-MIAS	Accuracy: 96.92%

**Table 2 biomedicines-11-01536-t002:** Semantic Segmentation Performance Comparison.

Models	Global Accuracy	Mean Accuracy	Mean IoU	Weighted IoU *	Weighted F1-Score	Parmas(KB)
Unet3+	0.909974	0.825364	0.525732	0.891479	0.936341	26359
Unet	0.907113	0.483172	0.367661	0.868647	0.925893	16858
FCN 32s	0.903111	0.511716	0.388243	0.866195	0.914389	18206
SegNet	0.890970	0.510140	0.330976	0.867940	0.913253	15931
DeepLabV3+	0.913325	0.648982	0.464559	0.874940	0.923186	39403
pspNet	0.901890	0.626284	0.460464	0.863611	0.911476	3992

* IoU (Intersection over union).

## Data Availability

The datasets produced and analyzed in this study are not publicly accessible due to IRB and institutional limitations.

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
