# Peer review of "Improving Breast Cancer Detection and Diagnosis through Semantic Segmentation Using the Unet3+ Deep Learning Framework"

_biomedicines, 2023, doi:10.3390/biomedicines11061536_

Round 1
Reviewer 1 Report
-The intro is good, but a medical/physiological overview of breast cancers needs to be articulated with strong supporting Refs
-A flowchart for the devised method, alongside an algorithmic table, needs to be fleshed out, explained and included
-The discussions require much more depth and synthesis in its depth
-Figure 10 needs improvement, and potential annotations and descriptive texts
-The conclusion needs a broader reflective overview, with a clear pathway for further work
n/a
Author Response
Point 1: The intro is good, but a medical/physiological overview of breast cancers needs to be articulated with strong supporting Refs
Response 1: Thank you for your worthy consideration and highlighting that particular point, we have added a brief pathophysiology of breast cancer in introduction part from line 30 - 40 (yellow highlighted) with strong references.
Point 2: A flowchart for the devised method, alongside an algorithmic table, needs to be fleshed out, explained and included.
Response 2: Thank you very much for pointing this out, we have designed the algorithm and workflow in figure 1 and explained in from line 82 – 92 (yellow highlighted).
Point 3. The discussions require much more depth and synthesis in its depth
Response 3: Thank you underpinning this point, we have modified the discussion part in depth (yellow highlighted).
Point 4. Figure 10 needs improvement, and potential annotations and descriptive texts
Response 4: Thank you for the comments, we have updated the asked figure with current assigned number as 6th and modified with the annotations and descriptive texts yellow highlighted).
Point 5. The conclusion needs a broader reflective overview, with a clear pathway for further work
Response 5: Many thanks for correction, we have updated the Conclusion part and mentioned its prospects (yellow highlighted).
Reviewer 2 Report
This paper presents a comparative analysis of multiple deep learning-based segmentation models for identifying and diagnosing breast cancer using ultrasound images. The topic is of interest for biomedical studies and the reported experimental results look interesting, but the manuscript should be further polished before being considered for publication. Here are a couple of suggestions and questions:
- The writing of the paper could be further improved for better clarity, e.g., in line 190, the authors should briefly summarize the method rather than simply referring to it as "this technique".
- When comparing the different models, is it guaranteed that each model has attained its optimum and the comparison is fair? If so, how?
- To gain a better view of this field, it could be helpful to add a few lines to compare different variants of deep learning schemes for diagnosis using ultrasound images, e.g., "Multi-instance Deep Learning with Graph Convolutional Neural Networks for Diagnosis of Kidney Diseases Using Ultrasound Imaging" in MICCAI-CLIP'19.
The writing is good in general but the clarity could be further improved.
Author Response
Point 1: The writing of the paper could be further improved for better clarity, e.g., in line 190, the authors should briefly summarize the method rather than simply referring to it as "this technique".
Response 1: Thanks for pointing it out, Methods has been discussed in the detailed in the introduction part from line 79-87 (yellow highlighted) .
Point 2: When comparing the different models, is it guaranteed that each model has attained its optimum and the comparison is fair? If so, how?
Response 2: Many thanks for the detailed comments, we have modified the Introduction part. In addition, as each model used same data set to train and test the algorithm, after comparing the results of all models with Unet3+ model, we analyzed that UNet3+ model is a state-of-the-art semantic segmentation architecture that showed optimal performance with an average accuracy of 82.53% and an average intersection over union (IU) of 52.57%, weighted IU 89.14% among all other models. However, DeeplabV3+ showed the better global accuracy (91.33%) comparing to Unet3+ (90.18%) and all other models.
Point 3. - To gain a better view of this field, it could be helpful to add a few lines to compare different variants of deep learning schemes for diagnosis using ultrasound images, e.g., "Multi-instance Deep Learning with Graph Convolutional Neural Networks for Diagnosis of Kidney Diseases Using Ultrasound Imaging" in MICCAI-CLIP'19.
Response 3: Thanks for the highlighting it out,we have discussed about different variants of deep learning for diagnosis using ultrasound images. From line 75-85.in the introduction part.